# Bridge the Points: Graph-based Few-shot Segment Anything Semantically

**Anqi Zhang[1], Guangyu Gao[1],[*] Jianbo Jiao[2], Chi Harold Liu[1], and Yunchao Wei[3]**

[1]School of Computer Science, Beijing Institute of Technology
[2]The MIx group, School of Computer Science, University of Birmingham
[3]WEI Lab, Institute of Information Science, Beijing Jiaotong University
`andy_zaq@outlook.com`

## Abstract

The recent advancements in large-scale pre-training techniques have significantly enhanced the capabilities of vision foundation models, notably the Segment Anything Model (SAM), which can generate precise masks based on point and box prompts. Recent studies extend SAM to Few-shot Semantic Segmentation (FSS), focusing on prompt generation for SAM-based automatic semantic segmentation. However, these methods struggle with selecting suitable prompts, require specific hyperparameter settings for different scenarios, and experience prolonged one-shot inference time due to the overuse of SAM, resulting in low efficiency and limited automation ability. To address these issues, we propose a simple yet effective approach based on graph analysis. In particular, a Positive-Negative Alignment module dynamically selects the point prompts for generating masks, especially uncovering the potential of the background context as the negative reference. Another subsequent Point-Mask Clustering module aligns the granularity of masks and selected points as a directed graph, based on mask coverage over points. These points are then aggregated by decomposing the weakly connected components of the directed graph in an efficient manner, constructing distinct natural clusters. Finally, the positive and overshooting gating, benefiting from graph-based granularity alignment, aggregate high-confident masks and filter out the false-positive masks for final prediction, without relying on additional hyperparameters and redundant mask generation. Extensive experimental analysis across tasks including the standard FSS, One-shot Part Segmentation, and Cross Domain FSS validate the effectiveness and efficiency of the proposed approach, surpassing state-of-the-art generalist models with a mIoU of 58.7% on COCO-20[i] and 35.2% on LVIS-92[i]. The project page of this work is: https://andyzaq.github.io/GF-SAM/.

## 1  Introduction

Previous semantic segmentation methods [1–8], which rely on the pixel-level classification, often struggle with generalization and overfitting due to limited labeled data. In addition, recent approaches, such as MaskFormer [9], have shifted the paradigm to mask-based classification, offering a more flexible approach to improving the segmentation performance by exploiting the consistency and completeness of generated class-agnostic masks. The Segment Anything Model (SAM) [10] further marks a significant advancement by utilizing extensive pre-training on huge-scale dataset SA-1B to achieve more robust, class-agnostic segmentation capabilities. SAM excels in producing precise masks across various domains using simple prompts such as points, boxes, and coarse masks. While the boundaries of these masks can closely align with object boundaries, the lack of semantic

---

[*]Corresponding Author.

38th Conference on Neural Information Processing Systems (NeurIPS 2024).

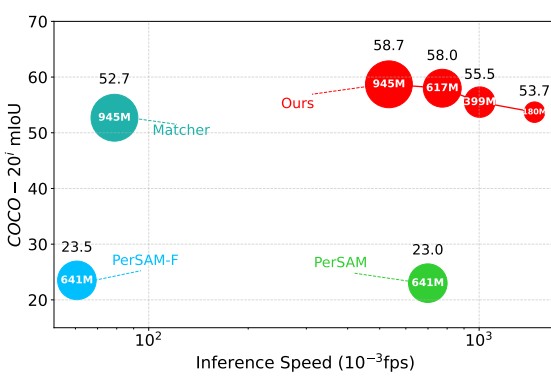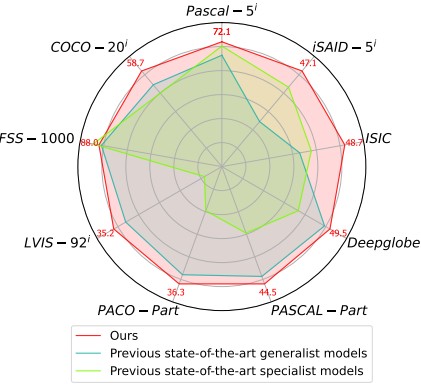

(a) Performance-efficiency comparison of FSS models. The numbers inside the points represent the numbers of parameters.

(b) Comparison with previous generalist and specialist models on various FSS datasets.

Figure 1: Performance comparisons of our approach against previous state-of-the-art methods regarding efficiency and generalized capabilities in Few-shot Semantic Segmentation. Figure 1(a) illustrates our approach's superior performance in efficiency and effectiveness across various model sizes. Figure 1(b) demonstrates the generalizability of our approach across different domains.

understanding and the requirement for manual prompts prevent SAM from being used in automatic semantic segmentation applications.

Recent studies have attempted to automate this process in the Few-shot Semantic Segmentation (FSS), using a few reference images and a fine-grained external backbone network (*e.g.*, DINOv2 [11]) to guide SAM in segmenting target semantic objects. However, these methods face two main challenges: achieving suitable points for precise and full coverage of the target object, and handling the ambiguity of SAM-generated masks, from partial to complete coverage. Specifically, they either utilize the most similar candidate point prompts for iterative mask generation and refinement [12], or build a restrictively selected set of point prompts for heuristically weighted mask merging based on manually designed metrics [13], outperforming both previous specialist methods [14–19] and generalist methods without SAM [20, 21]. However, these methods overlooked the underlying relationships between points (derived from fine-grained features) and masks (generated by SAM in a coarse-grained manner). This oversight led to low efficiency (as indicated in Fig. 1(a)) and limited automation capabilities. Alignment between these two types of granularity could uncover the potential of simple decision-making on masks, which can eliminate redundant refinement and manual hyperparameter selection for complicated metrics.

In this paper, we explicitly explore the relationship between point prompts and corresponding masks from SAM, and present a simple yet effective parameter-free framework with only one-time mask generation to segment anything semantically, in a graph-based few-shot manner. We first introduce a Positive-Negative Alignment (PNA) module to dynamically select point prompts using foreground and background references from reference images. Unlike existing methods, our approach combines different granularity by constructing a directed graph according to mask coverage over points. Then, we perform connectivity analysis on the constructed graph to obtain several weakly connected components as automatic clustering of point prompts, which bridges points and masks as well as fine-grained and coarse-grained features. To mitigate the inevitable introduction of false positives in the PNA module, we further leverage weakly connected component clusters and limited semantic information in selected points, to more accurately filter and merge masks that mismatch in different granularities. In particular, our proposed method involves two post-gating based on weakly connected clusters: the positive gating retains masks capturing a greater proportion of potential target areas, while the overshooting gating screens out outlier points near object boundary, with coverage self-consistency consideration.

Extensive experimental analysis on Few-shot Semantic Segmentation demonstrates both the efficiency and effectiveness of our approach, as shown in Fig. 1(b). We first conduct the experiments on generalized FSS datasets, including Pascal-5[i] [22], COCO-20[i] [23], FSS-1000 [24] and LVIS-92[i] [13]. Our approach surpasses existing state-of-the-art approaches on these datasets, with $5.8\%$ and $2.2\%$

of improvement respectively on more challenging COCO-20$^i$ and LVIS-92$^i$. As for the challenging One-shot Part Segmentation, our approach still exceeds previous methods with 1.6% of mIoU on both PACO-Part and PASCAL-Part. Furthermore, to demonstrate the ability of our approach across different domains, we perform an evaluation on several specific datasets, including Deepglobe [25], ISIC [26], and iSAID-5$^i$ [27]. The proposed approach establishes new state-of-the-art performance on mIoU with 49.5% on Deepglobe, 48.7% on ISIC, and 47.3% on iSAID-5$^i$.

Overall, our contributions are summarized as follows:

- We present, to our knowledge, the first graph-based approach for SAM-based few-shot semantic segmentation, modeling the relationship of SAM-generated masks in an automatic clustering manner.
- We propose a positive-negative alignment module and a post-gating strategy based on the weakly connected graph components, enabling a hyperparameter-free pipeline.
- Extensive experimental comparisons and analysis across several datasets over various settings show the effectiveness and efficiency of the proposed method.

## 2 Related Work

**Few-shot semantic segmentation.** Few-shot Semantic Segmentation (FSS) [22] aims to segment the target object using only a limited number of annotated reference samples for guidance. Previous FSS methods are mainly categorized into two types, namely the methods based on prototype matching [28–34] and methods based on pixel-wise matching [35–40]. The methods based on prototype matching, *e.g.* PFENet [31], BAM [41], SSP [42], use the Mask Average Pooling operation from SGOne [43] to generate a prototype as a global representation of the reference features, and compare the target features with the prototypes. The methods based on pixel-wise matching compute the correlation of all pixels between target and reference features. Then different methods address the correlations through distinct mechanisms, such as 4D Convolution (*e.g.*, HSNet [14]) and Transformer (*e.g.*, HDMNet [44], AMFormer [45]). Although these specialist models perform significantly on specific tasks, they are prone to overfitting the training samples and often struggle to adapt to domain shifts.

**SAM-based semantic segmentation.** Recently, Segment Anything Model (SAM) [10] has shown remarkable zero-shot class-agnostic segmentation capabilities using prompts like points, boxes, and coarse masks. However, the coarse-grained feature representation of SAM limits its effectiveness for fine-grained semantic segmentation tasks. Several approaches have been proposed to extend SAM for semantic segmentation. For example, Semantic-SAM [46] jointly train the model on SA-1B and other semantic aware segmentation datasets to enhance granularity. OV-SAM [47] combines SAM and CLIP [48] for open-vocabulary semantic segmentation. Moreover, some methods introduce SAM into FSS tasks. PerSAM and PerSAM-F [12] leverage SAM for personalized segmentation with one-shot guidance. Matcher [13] uses a SAM-based training-free structure, achieving impressive performance in both FSS and One-shot Part Segmentation. VRP-SAM [49] trains an external Visual Reference Prompt Encoder to automatically generate prompts from reference images using points, scribble, box, or masks. However, previous training-free methods struggled to balance performance and efficiency, often relying on excessive external manual hyperparameters.

## 3 Preliminaries

Few-shot Semantic Segmentation (FSS) aims to segment target objects in an image with a few annotated reference images. Consider a scenario where each group of samples contains a target image $x^t$ and a reference image set $R = \{x_k^r, y_k^r\}_{k=1}^K$ with the size of $H \times W$, where $x_k^r$ and $y_k^r$ mean the $k_{th}$ reference image and its corresponding mask. Focusing on the 1-shot case, where $K = 1$, it begins with a feature extraction backbone network $f_B(\cdot)$, which encodes both $x^t$ and $x^r$ into semantic features $F^t$ and $F^r$ in $\mathbb{R}^{hw \times c}$, where $h$ and $w$ denote the height and width of the feature maps, and $c$ is the feature dimension. Subsequent few-shot processes utilize these feature maps to generate a predicted segmentation $\tilde{y} \in \mathbb{R}^{H \times W}$ for $x^t$. This prediction is then compared to the Ground Truth (GT) $y^t$ for evaluation.

The Segment Anything Model (SAM) is a generalized foundation segmentation model adept at generating precise masks based on varied prompts of points, boxes, and coarse masks. Built around

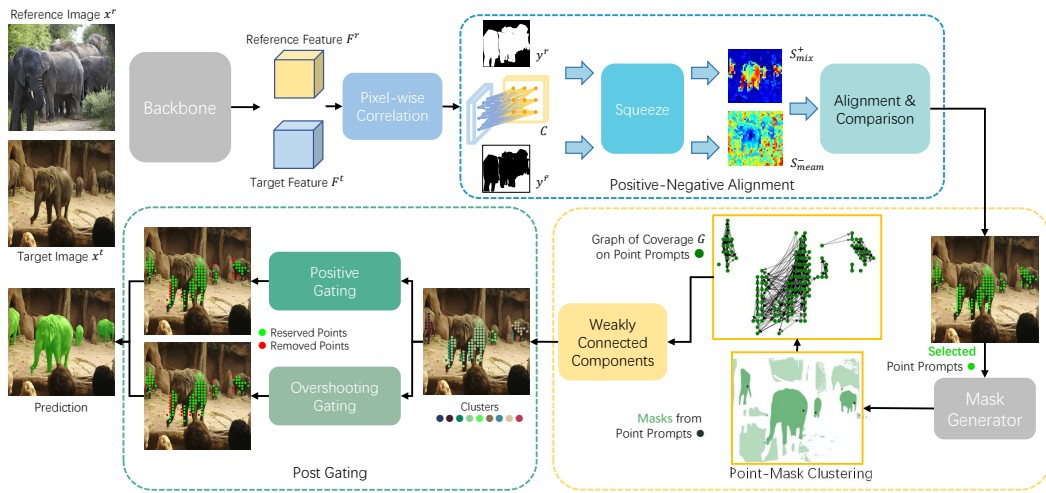

Figure 2: Overview of our approach, where the Positive-Negative Alignment module recognizes the correlation between target features and reference features for point selection, the Point-Mask Clustering module efficiently clusters the points based on the coverage of corresponding masks, and Post-Gating filters out the false-positive masks for generating final prediction.

a core architecture that includes an image encoder, a prompt encoder, and a mask decoder, SAM effectively processes input images $x^t$ and prompts $P$ to produce detailed segmentation masks $\hat{y}$. These masks accurately delineate specific objects or regions within the images, based on the guidance provided by the prompts.

## 4 Method

Diverging from traditional methods, we use a directed graph to exploit the natural relationships between points and their corresponding masks, representing fine-grained and coarse-grained features, respectively. As shown in Fig. 2, our approach mainly comprises the Positive-Negative Alignment (PNA) module, Point-Mask Clustering (PMC) module, and Post-Gating strategy. The PNA module leverages semantic features from the backbone network to sort pixel-wise correlations into similarity maps, enabling precise point selection. The PMC module then clusters masks based on these selected points, while Post-Gating strategy refines the selection, enhancing the accuracy and reliability of the final prediction.

### 4.1 Positive-Negative Alignment for Point Selection

The PNA module efficiently selects point prompts to balance the number of points and coverage of target objects. Using the semantic features $F^r$ and $F^t$ from the reference and target images respectively (with *e.g.*, DINOv2 [11]), we get the pixel-wise correlation matrix $C \in \mathbb{R}^{hw \times hw}$:

$$C(i,j) = ReLU\left(\frac{F^t(i) \cdot F^r(j)}{\|F^t(i)\| \cdot \|F^r(j)\|}\right), \tag{1}$$

where $C(i,j)$ represents the similarity between the $i$-th pixel of target features $F^t(i)$ and the $j$-th pixel of reference features $F^r(j)$.

To minimize hyperparameter reliance, we leverage background features typically overlooked in FSS, indicated by the negative mask $y^{\tilde{r}} = \neg y^r$ of the reference image. According to $y^r$ and $y^{\tilde{r}}$, we divide $C$ into $C^+$ and $C^-$ in $\mathbb{R}^{hw \times hw}$ for foreground and background features, respectively. We then introduce two positive similarity maps in mean and max aspects respectively:

$$S^+_{mean}(i) = \frac{\sum_{j=1}^{hw} C^+(i,j)}{\sum_{j=1}^{hw} \mathcal{I}(y^r)_j}, \quad S^+_{max}(i) = max(C^+(i)), \tag{2}$$

where $\mathcal{I}$ resizes $y^r$ to the same resolution as $F^r$ and then flatten it into a vector, $max(\cdot)$ finds the maximum value in the $i$-th row of $C^+$. The mean positive similarity map $S^+_{mean} \in \mathbb{R}^{hw}$ captures global similarity towards the reference object but may blur distinct internal features, reducing accuracy for complex objects. In contrast, the max positive similarity map $S^+_{max} \in \mathbb{R}^{hw}$ focuses on the most similar regions, enhancing recall but also increasing noise. To maintain distinctiveness while reducing noise, we introduce the mixture similarity map $S^+_{mix} = S^+_{mean} \odot S^+_{max}$ using the Hadamard product. This method boosts target region distinctiveness by merging the strengths of both maps, while diminishing noise through the more stable global similarity.

To select prompt points, we also use the mean negative similarity map $S^-_{mean}$, which reflects background similarity, noting that similar objects typically share higher background similarity values. We then align $S^+_{mix}$ and $S^-_{mean}$ by min-max normalization $\mathcal{M}$ to get:

$$S_p(i) = \mathcal{M}(S^+_{mix})(i) \cdot \mathbf{1}_{\{\mathcal{M}(S^+_{mix})(i) > \mathcal{M}(S^-_{mean})(i)\}}, \tag{3}$$

where $S_p \in \mathbb{R}^{hw}$ is the filtered map for point selection, and $\mathbf{1}_{\{\cdot\}}$ is 1 if the condition is true and 0 otherwise. Although we minimize false negatives, noise points remain. To select suitable points from $S_p$ without hyperparameters, we define the sum of elements in $S_p$ as the number $N$ of points to be selected. We then pick the $N$ highest-value points from $S_p$ as the point prompt set $\boldsymbol{P} = \{P_l\}^N_{l=1}$.

## 4.2 Point-Mask Clustering with Graph Connectivity

We utilize point prompts from $\boldsymbol{P}$ to generate masks with SAM. Each point $P_l$ in $\boldsymbol{P}$ corresponds to a unique mask $\hat{y}_l \in \mathbb{R}^{H \times W}$. As our point selection strategy prioritizes the coverage of objects, false-negative masks are unavoidable. Moreover, mask coverage can vary significantly within the same region, ranging from partial to full object coverage. This necessitates understanding the internal relationships among coarse-grained masks and points from fine-grained feature comparison to ensure those covering the same target are accurately gathered.

To address this, we design the Point-Mask Clustering (PMC) module, which clusters points and their corresponding masks based on mask coverage over points. Following the principles of efficiency and automation, the PMC module is based on a directed graph $G = (V, E)$ with the vertex $v_l$ in $V$ representing point $P_l$ and its corresponding mask $\hat{y}_l$. Edges in $E$ are established based on mask coverage over other points; an edge $e_{l,m}$ exists if mask $\hat{y}_l$ covers points $P_m$ (with $m \neq l$). Specifically, we do not establish edges for masks covering their corresponding points to avoid creating loops.

The graph $G$ is a directed simple graph, allowing us to cluster vertices by identifying weakly connected components. This clustering process is hyperparameter-free, ensuring that every pair of vertices $u, v \in V$ within the same component has a directed path between them. Each weakly connected component encompasses a set of points $\hat{P}_p$ (with $P = \{\hat{P}_p\}$) that are all covered by the union of their masks in $\hat{M}_p$, where $p$ indexes the clusters.

The advanced SAM plays a crucial role in maintaining the precision of the generated masks. The precision of high-quality masks typically ensures non-overlapping between masks and prompting points of adjacent regions, especially those of different categories. This is the precondition for the efficacy of our PMC module, as even slight errors could significantly impact the clustering accuracy.

## 4.3 Post-Gating on Weakly Connected Components

Our PNA module, while efficient in selecting points, inadvertently includes false positives, as detailed in Sec. 4.1. To mitigate this, we further develop two gating strategies targeting distinct types of false positives based on clusters formed from weakly connected components.

**Positive gating.** Despite the method in Sec. 4.1 diminishing the noise points outside the target region, there are still a few remaining noise points. These issues may have minimal impact on traditional segmentation methods, but under the SAM framework, masks derived from these noise points can significantly degrade accuracy. Moreover, some clusters of masks may extend beyond their intended target regions due to inaccuracies in SAM-generated masks or because the targeted object is part of a larger entity. Thus, we propose a Positive Gating strategy to address these issues.

This strategy prioritizes mask effectiveness by assessing whether a mask contains more positive than negative pixels, thereby facilitating a specialized designed mask growth for final prediction. The

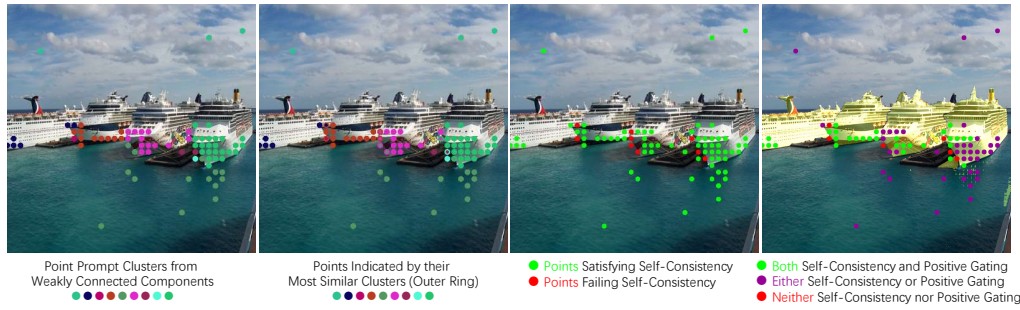

Point Prompt Clusters from
Weakly Connected Components

Points Indicated by their
Most Similar Clusters (Outer Ring)

● Points Satisfying Self-Consistency
● Points Failing Self-Consistency

● Both Self-Consistency and Positive Gating
● Either Self-Consistency or Positive Gating
● Neither Self-Consistency nor Positive Gating

Figure 3: Illustration of the Overshooting Gating strategy. The outer ring of points in the second image indicates the most similar cluster of corresponding points, *i.e.*, points with different outside and inside colors do not satisfy the self-consistency.

focus of mask growth is to enhance coverage of the target area rather than multiple objects, while minimizing background inclusion. Firstly, this method utilizes a parameter-free gating mechanism that discriminates between pixel polarities, based on the positive and negative similarity maps, $S_{mean}^+$ and $S_{mean}^-$, as described in Sec. 4.1. To achieve this, we utilize $S_{mean}^+$ and $S_{mean}^-$, along with the median of $S_{mean}^+$ (*i.e.*, the midpoint between the maximum and minimum values of $S_{mean}^+$), to constructs the polarity map $\bar{R}$ as follows:

$$\bar{R}(i) = \begin{cases} 1, & S_{mean}^+(i) \times S_{mean}^+(i) > s_{mid} \times S_{mean}^-(i), \\ -1, & else. \end{cases} \tag{4}$$

Then, using the polarity map $\bar{R}$, we calculate the number of positive pixels of the $l^{th}$ mask as follows:

$$s_l^+ = \sum_{i=1}^{hw} \bar{R}(i) \odot \mathcal{I}(\hat{y}_l)(i), \tag{5}$$

where $\mathcal{I}$ resizes and flattens $\hat{y}_l$ to the feature map dimensions. Subsequently, for each cluster $\hat{M}_p$ of weakly connected components, we sort the masks according to the ratio of positive pixel numbers to their areas. The indices of these sorted masks are denoted by $Q$. We then initialize a blank pseudo mask $\ddot{y}_p \in \mathbb{R}^{H \times W}$ and a set of positive points $P^+$. Following this, we apply a Mask Growth algorithm as outlined in Sec. A.1 and Alg. 1 for maintaining positive masks. This algorithm iteratively evaluates whether the region of $\hat{y}_q$ outside the pseudo mask $\ddot{y}_p$ is positive, updates $\ddot{y}_p$ with the identified positive mask, and adds its corresponding point into $P^+$.

**Overshooting gating.** The fine-grained semantic features from $f(\cdot)$ are reliable for locating target objects, yet the point coverage of the target areas varies, leading to both under-coverage and over-coverage. SAM effectively addresses under-coverage; however, over-coverage, which extends beyond target boundaries, often produces false-positive masks. These overshooting points, while semantically similar to the target areas in $F^t$, typically derive masks that cover areas outside the target, resulting in a mismatch of representations between the granularity of points and masks. Thus, these points cannot be clustered with points inside the target areas.

Hence, we devise an overshooting gating strategy with consideration of self-consistency to eliminate overshooting points and their associated masks. As shown in Fig. 3, We assess the similarity between the features of each point $P_l$ and the union mask $\hat{y}_p \in \mathbb{R}^{H \times W}$ from each mask cluster $\hat{M}_p$. The similarity computation for estimating self-consistency is performed as follows:

$$s^{sc}(l, p) = \frac{\sum_{i=1}^{hw} Sim(F^t(P_l), (F^t \odot \mathcal{I}(\hat{y}_p))(i))}{\sum_{i=1}^{hw} \mathcal{I}(\hat{y}_p) \cdot dist(l, p)}, \tag{6}$$

where $Sim(\cdot, \cdot)$ refers to the correlation calculation mentioned in Eq. 1. We introduce an external function $dist(\cdot, \cdot)$ to measure the distance in $F^t$ between each point $P_l$ and the nearest selected point in $\hat{P}_p$. This measure helps confine comparison to neighboring clusters, minimizing interference from other instances. We then identify the cluster most similar to the points and retain those in the point set $P^{sc}$ that are more similar to their respective clusters.

Table 1: Performance on Few-shot Semantic Segmentation datasets of Pascal-5$^i$, COCO-20$^i$, FSS-1000, and LVIS-92$^i$. Gray means the in-domain trained results. The best results are shown in **bold**.

| Methods | Pascal-5$^i$ 1-shot | Pascal-5$^i$ 5-shot | COCO-20$^i$ 1-shot | COCO-20$^i$ 5-shot | FSS-1000 1-shot | FSS-1000 5-shot | LVIS-92$^i$ 1-shot | LVIS-92$^i$ 5-shot |
|---|---|---|---|---|---|---|---|---|
| *specialist model* | | | | | | | | |
| HSNet [14][CVPR21] | 66.2 | 70.4 | 41.2 | 49.5 | 86.5 | 88.5 | 17.4 | 22.9 |
| VAT [50][ECCV22] | 67.9 | 72.0 | 41.3 | 47.9 | 90.3 | 90.8 | 18.5 | 22.7 |
| HDMNet [44][CVPR23] | 69.4 | 71.8 | 50.0 | 56.0 | - | - | - | - |
| AMFormer [45][NeurIPS23] | 70.7 | 73.6 | 51.0 | 57.3 | - | - | - | - |
| *generalist model* | | | | | | | | |
| PerSAM [12][ICLR24] | 43.1 | - | 23.0 | - | 71.2 | - | 11.5 | - |
| PerSAM-F [12][ICLR24] | 48.5 | - | 23.5 | - | 75.6 | - | 12.3 | - |
| Matcher [13][ICLR24] | 68.1 | 74.0 | 52.7 | 60.7 | 87.0 | **89.6** | 33.0 | 40.0 |
| VRP-SAM [49][CVPR24] | 71.9 | - | 53.9 | - | - | - | - | - |
| Ours | **72.1** | **82.6** | **58.7** | **66.8** | **88.0** | 88.9 | **35.2** | **44.2** |

Table 2: Performance on One-shot Part Segmentation datasets and Cross Domain Few-shot Semantic Segmentation datasets. The best results are shown in **bold**.

| Methods | One-shot Part Seg. PASCAL-Part 1-shot | One-shot Part Seg. PACO-Part 1-shot | Cross Domain FSS Deepglobe 1-shot | Cross Domain FSS Deepglobe 5-shot | Cross Domain FSS ISIC 1-shot | Cross Domain FSS ISIC 5-shot | Cross Domain FSS iSAID-5$^i$ 1-shot | Cross Domain FSS iSAID-5$^i$ 5-shot |
|---|---|---|---|---|---|---|---|---|
| *specialist model* | | | | | | | | |
| HSNet [14][CVPR21] | 32.4 | 22.6 | 29.7 | 35.1 | 31.2 | 35.1 | 34.1 | 40.4 |
| DRA [51][CVPR24] | - | - | 41.3 | 50.1 | 40.8 | 48.9 | - | - |
| FRINet [52][TGRS23] | - | - | - | - | - | - | 42.6 | 44.5 |
| *generalist model* | | | | | | | | |
| PerSAM [12][ICLR24] | 32.5 | 22.5 | 31.4 | - | 23.9 | - | 19.2 | - |
| PerSAM-F [12][ICLR24] | 32.9 | 22.7 | 35.0 | - | 23.6 | - | 20.3 | - |
| Matcher [13][ICLR24] | 42.9 | 34.7 | 48.1 | 50.9 | 38.6 | 35.0 | 33.3 | 34.3 |
| Ours | **44.5** | **36.3** | **49.5** | **57.7** | **48.7** | **55.2** | **47.1** | **52.4** |

**Mask Merging.** Finally, we obtain two distinct sets of points, namely $P^+$ and $P^{sc}$. We then union the masks corresponding to points that are common to both $P^+$ and $P^{sc}$. The merged masks form the final prediction, denoted as $\tilde{y}$.

# 5 Experimental Results

## 5.1 Datasets

To illustrate the Few-shot Semantic Segmentation ability and generalization capacity, we conduct three types of sub-tasks, *i.e.* standard Few-shot Semantic Segmentation, One-shot Part Segmentation, and Cross Domain Few-shot Semantic Segmentation. The datasets for these tasks are as follows:

**Pascal-5$^i$**, **COCO-20$^i$**, **FSS-1000**, and **LVIS-92$^i$** are standard FSS datasets. Pascal-5$^i$ [22] is based on the Pascal VOC 2012 [53] and SDS [54]. The 20 classes are separated into 4 folds of 5 classes. COCO-20$^i$ [23] is an 80-class dataset from MSCOCO [55], which has 4 folds with each fold containing 20 classes. FSS-1000 [24] contains 1000 classes. The training, validation, and testing folds contain 520, 240, and 240 classes, respectively. LVIS-92$^i$ [13] is more challenging for evaluating generalist models, which select 920 classes with more than 2 images and divide these classes into 10 folds.

**PASCAL-Part** and **PACO-Part** [13] are One-shot Part Segmentation datasets. PASCAL-Part [56, 57] contains 56 different object parts in 4 superclasses. PACO-Part is built based on the PACO dataset [58], which has 456 object part classes. The 303 classes with at least 2 samples in PACO-Part are divided into four folds following Matcher [13].

**Deepglobe**, **ISIC2018**, and **iSAID-5$^i$** are Cross Domain FSS datasets. The Deepglobe [25] contains satellite images of geographic categories including urban, agriculture, rangeland, forest, water, and

Table 3: Ablation study of Point Selection.

| $S^+_{mean}$ | $S^+_{max}$ | $S^-_{mean}$ | Top $N$ | mIoU |
|:---:|:---:|:---:|:---:|:---:|
| ✓ | | ✓ | ✓ | 53.1 |
| | ✓ | ✓ | ✓ | 54.1 |
| ✓ | ✓ | ✓ | | 56.4 |
| ✓ | ✓ | | ✓ | 51.5 |
| ✓ | ✓ | ✓ | ✓ | **58.7** |

Table 4: Ablation study of PMC and Post-Gating.

| PG | | OG | | COCO-20i | LVIS-92i |
|:---:|:---:|:---:|:---:|:---:|:---:|
| Strong | Weak | Strong | Weak | | |
| | ✓ | | | 44.0 | 24.2 |
| ✓ | | | | 57.1 | 34.3 |
| ✓ | | | | 57.1 | 33.9 |
| ✓ | | ✓ | | 56.7 | **35.2** |
| ✓ | | | ✓ | **58.7** | **35.2** |
| k-means++ | | | | 57.5 | 34.0 |

Table 5: Ablation study of positive gating on each cluster. M.G. represents the Mask Growth algorithm.

| Strategies | M.G. | COCO-20i | PASCAL-Part |
|:---:|:---:|:---:|:---:|
| Sum | | 55.3 | 39.1 |
| | ✓ | 58.6 | 44.3 |
| Num | | 57.1 | 42.2 |
| | ✓ | **58.7** | **44.5** |

Table 6: Ablation on the strategies of Self-Consistency measurement.

| Strategies | mIoU | Δ |
|:---:|:---:|:---:|
| None | 57.1 | 0.0 |
| Point Sim. | 56.7 | -0.4 |
| MAP Sim. | 57.7 | +0.6 |
| Mean Sim. W/o dist | 49.1 | -8.0 |
| Mean Sim. (Ours) | **58.7** | **+1.6** |

barren. The ISIC2018 [26] is a skin lesion analysis dataset with three classes. The iSAID-5i [27] evenly split 3 folds for 15 classes based on the remote sensing dataset iSAID [59].

## 5.2 Implementation Details

Following the settings of PerSAM [12] and Matcher [13] for a fair comparison, we use DINOv2 [11] with a ViT-L/14 [60] as our feature extraction backbone, and SAM [10] with ViT-H as the mask generator. The input image sizes are set to $518 \times 518$ for DINOv2 and $1024 \times 1024$ for SAM following Matcher [13]. Except for the default hyperparameters of SAM and DINOv2, our approach **does not have any external hyperparameter**. We apply the mean Intersection over Union (mIoU) metric for evaluating the performance. All experiments are conducted on a single NVIDIA RTX2080Ti.

## 5.3 Comparison with State-Of-The-Arts

**Comparison on the standard FSS datasets.** We compared our approach with other state-of-the-art specialist and generalist models. As shown in Tab. 1, our approach achieves 72.1% mIoU on the Pascal-5i dataset and 58.7% mIoU on COCO-20i dataset, which surpasses all previous specialist and generalist state-of-the-art models. Our approach reaches 35.2% mIoU in the more challenging dataset of LVIS-92i, with 2.2% of improvement compared to the previous training-free method Matcher. The performance remains competitive on the FSS-1000 compared with specialist models. The 5-shot result of Pascal-5i, COCO-20i, and LVIS-92i further extends the lead, which shows that our approach can effectively handle the few-shot scenario.

**Comparison on the One-shot Part Segmentation datasets.** The One-shot Part Segmentation tasks evaluate the ability to fetch the target part from the whole object. The results in Tab. 2 show that our approach achieves the mIoU of 44.5% and 36.3% on both datasets of PASCAL-Part and PACO-Part, respectively. Our approach outperforms the state-of-the-art generalist model Matcher with 1.6% on both datasets. Given that Matcher employs **specific hyperparameters** to enhance part segmentation, our superior performance demonstrates the adaptability of our approach across both object and part segmentation contexts.

**Comparison on the Cross Domain FSS datasets.** The Cross Domain FSS tasks validate the performance on different domains. Our approach achieves state-of-the-art performance in datasets of Deepglobe, ISIC, and iSAID-5i among other specialist domain models and generalist models. Especially within the context of the skin lesion analysis dataset ISIC and remote sensing dataset iSAID-5i, our approach outperforms Matcher by margins of 10.1% and 13.8% respectively.

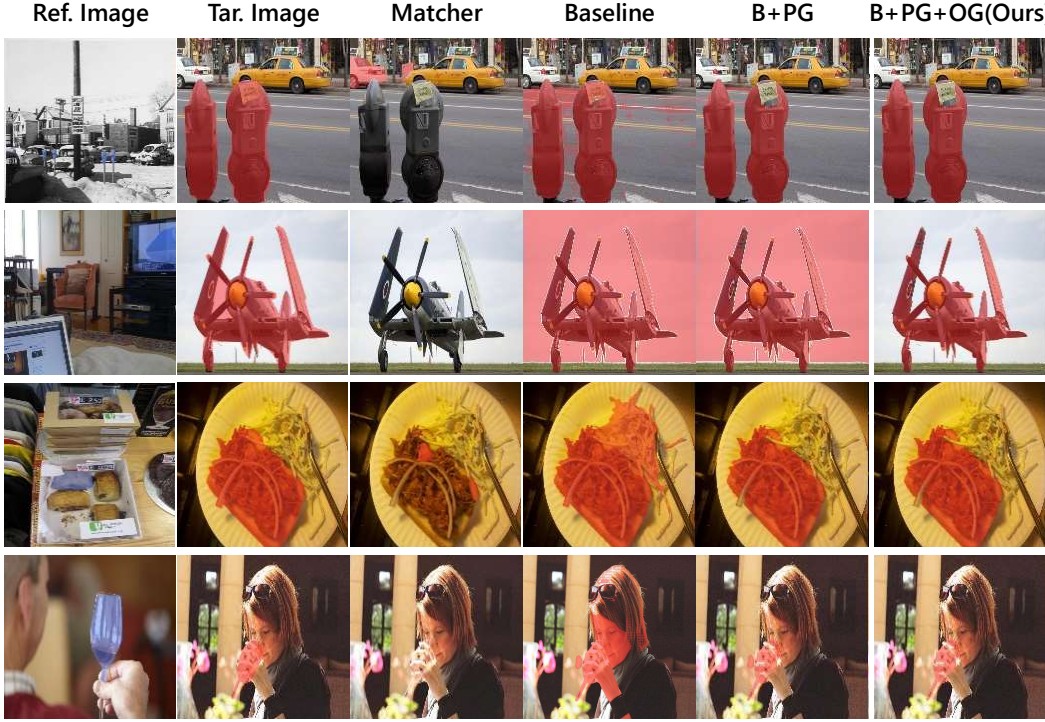

| Ref. Image | Tar. Image | Matcher | Baseline | B+PG | B+PG+OG(Ours) |

Figure 4: Qualitative analysis of Matcher, Baseline, B+PG, B+PG+OG. B, PG, and OG respectively represent Baseline, Positive Gating, and Overshooting Gating. Masks in ref. image are shown in blue.

## 5.4 Ablation Study

**Point selection.** We evaluate the impact of various similarity maps and the parameter-free selection of top N points on performance, as detailed in Sec. 4.1. As shown in Tab. 3, using either $S^+_{mean}$ or $S^+_{max}$ alone leads to a performance drop of up to $5.6\%$ compared to using both. This decline is due to the inherent limitations of $S^+_{mean}$ and $S^+_{max}$ discussed in Sec 4.1. Additionally, the evaluation confirms that picking the top-N points based on similarity, which is parameter-free and requires no additional settings, simplifies the process and increases accuracy by $2.1\%$.

**Clustering method.** We compare our PMC module using weakly connected components with the PMC module using strong connected components, which provides finer clustering results. According to our experiment results in Tab. 4, the clusters from weakly connected components provide better performance on COCO-20[i] and LVIS-92[i] for both gating, as these clusters of masks have ideal coverage of the objects. Simply filtering the masks without clustering-based gating can only achieve $44.0\%$ mIoU on COCO-20[i] and $24.2\%$ mIoU on LVIS-92[i], which is significantly lower than the performance achieved with clustering-based gating. Furthermore, our dynamic hyperparameter-free clustering method outperforms the k-means++ with $1.2\%$ on both datasets. Note that k of k-means++ is set to 10 following Matcher [13].

**Positive gating.** Our approach compares the number of positive points and negative points in $\hat{S}^+$ (Num) to judge whether the mask is positive. We conduct experiments for the strategy of comparing the sum of positive and negative values (Sum). The results in Tab. 5 demonstrate the Num strategy yields better performance, as comparing the number of pixels mitigates the influence of a few excessively high similarity values. Furthermore, the utilization of the Mask Growth algorithm improves both FSS and Part Segmentation performance by carefully retaining the positive regions. However, it weakens the improvement of Num due to their similar effects.

**Overshooting gating.** Our Overshooting gating aims to filter out the overshooting points closely neighboring to target regions, thus having a less remarkable improvement of $1.6\%$ compared to Positive Gating, as shown in Tab. 6. This performance still surpasses the mean similarity of com-

paring points with regions of clustered points (Point Sim.) or the prototypes from Masked Average Pooling [43] with union masks (MAP Sim.). More importantly, the distance function avoids the gating from 9.6% of performance decline. It ensures each cluster only affects its neighboring points.

## 5.5 Qualitative Analysis

Here we present the qualitative results of Matcher, Baseline (1st row in Tab. 4), Baseline+PG (2nd row in Tab. 4) and our approach in Fig. 4. The bipartite matching of Matcher has a negative influence when the areas of the target object in reference and target images have significant differences, as shown in the 1st and 3rd rows. The positive gating with clustering filters out the noise masks in the 3rd row, while the Overshooting Gating further removes the masks belonging to overshooting points in the 2nd and 4th rows. More qualitative analyses please refer to the appendix.

## 6   Conclusion

In this paper, we proposed an efficient, training-free SAM-based FSS approach that requires no external hyperparameters. As an automatic SAM-based semantic segmentation pipeline, our approach balanced candidate points and object coverage in the Positive-Negative Alignment (PNA) module, then used SAM-generated masks in the Point-Mask Clustering (PMC) module to enhance Post Gating. Extensive experiments validated the superior performance of our approach, advancing semantic segmentation without extensive parameter tuning or training.

## Acknowledgments and Disclosure of Funding

This work was supported by the National Natural Science Foundation of China under No. 62472033, No. U23A20314, and No. 61972036. J. Jiao is supported by the Royal Society Short Industry Fellowship (SIF\R1\231009) and the Amazon Research Award.

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

# A   Appendix / supplemental material

## A.1   More Details for Mask Growth Algorithm

We mention the Mask Growth algorithm in Sec. 4.3. The Mask Growth algorithm is designed for each cluster of masks $\hat{M}_{weak,p}$. The details of the algorithm are shown in Alg. 1. We first initialize an empty set $P^+$ and a blank pseudo mask $\ddot{y}_p$. Then, we start an iterative process and get the current mask $\hat{y}_q$ based on the sorted sequence of indices $Q$. The parts of the current mask $\hat{y}_q$ overlapping with $\ddot{y}_p$ are removed. We compute the positive value $s_q^+$ of the remaining parts. If $s_q^+$ is positive, the mask $\hat{y}_q$ is updated into the $\ddot{y}_p$ and its corresponding point $P_q$ is added into $P^+$. As soon as the iterative process finishes, the set of positive points $P^+$ is established.

---

**Algorithm 1** Mask Growth for each cluster

---

**Input:** $\hat{M}_p, \ddot{y}_p, Q, P^+$
  **for** $n = 1$ to $|Q|$ **do**
    $q \leftarrow Q(n)$
    $\hat{y}_q \leftarrow \hat{M}_p(q)$
    $\hat{y}_q = \hat{y}_q \& \sim \ddot{y}_p$
    $s_q^+ \leftarrow \sum_{i=1}^{hw} \hat{S}^+(i) \odot \mathcal{I}(\hat{y}_q)(i)$
    **if** $s_q^+ > 0$ **then**
      Add $P_q$ to $P^+$.
      $\ddot{y}_p = \hat{y}_q \vee \ddot{y}_p$
    **end if**
  **end for**
**Output:** $P^+$

---

## A.2   Limitations

Our approach has impressive performance on Few-shot Semantic Segmentation tasks. However, due to the resolution of features $F^t$ from DINOv2 not aligning with the required resolution for prompting the SAM, we directly map the coordinates of points in $F^t$ to coordinates for prompting. This results in coordinate bias for small objects, as the gap between neighboring points can reach approximately 28 pixels. Our future work will focus on locating small objects.

## A.3   Societal Impacts

As a completely automatic SAM-based few-shot semantic segmentation approach without external hyperparameters, our method is capable of handling various scenarios of semantic segmentation, as demonstrated by our extensive experiments. The efficiency and generalizability of our method ensure a wide range of applications. Furthermore, since our training-free method is constructed upon the widely used open-source foundation models, we have not identified the negative societal impact to date.

## A.4   Details of Current SAM-based FSS Methods

Our approach aims to address several issues present in previous SAM-based FSS methods to achieve an automatic SAM-based model. These issues include the requirement of excessive external hyperparameters, overusing the mask generator of SAM, prolonged inference times, etc. The Tab. 7 shows the difference between our approach and previous SAM-based FSS methods. Fig. 5 shows the difference in using SAM as the mask generator between our approach and previous methods. Fig. 5(a) presents the iterative refinement of PerSAM, which involves generating masks from SAM 3 times. Fig. 5(b) exhibits that Matcher introduces an external Automatic Mask Generator, which automatically prompts for generating all mask proposals in the image. Our approach in Fig. 5(c) only utilizes the standard Mask Generator of SAM and generates the masks with our prompts only once.

Table 7: Details of the current SAM-based FSS methods.

| Methods | PerSAM | PerSAM-F | Matcher | VRP-SAM | Ours |
|---|---|---|---|---|---|
| Training-free | ✓ | | ✓ | | ✓ |
| External-hyperparameters-free | ✓ | | | | ✓ |
| Once mask generation | | | | ✓ | ✓ |
| Inference speed (s/img) | 1.43 | 16.5 | 12.7 | N/A | 1.88 |
| COCO-20$^i$ mIoU | 23.0 | 23.5 | 52.7 | 53.9 | 58.7 |

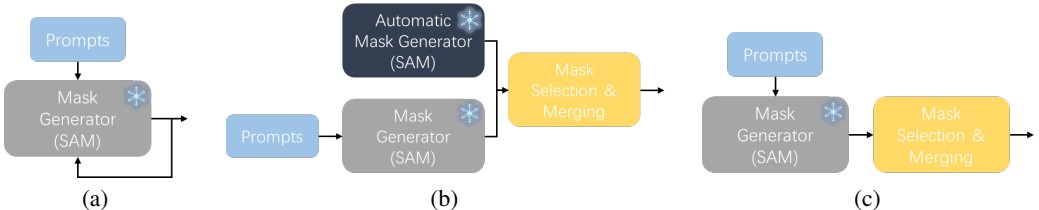

(a)  (b)  (c)

Figure 5: Comparison of the pipeline between the previous methods and our approach. (a) Per-SAM [12] iteratively uses the Mask Generator to refine the mask. (b) Matcher [13] introduced an external Automatic Mask Generator [10] with automatic prompting to excessively generate masks from the whole image. (c) The effectiveness of the PMC module and Post-Gating ensures that our approach uses Mask Generator with our prompts only once.

## A.5 Discussion of SAM

### A.5.1 Features from ViT Encoder of SAM

Previous state-of-the-art generalist FSS methods [] use DINOv2 or ResNet-50, instead of the default ViT encoder of SAM, for fine-grained features. We visualize the representative samples of Pascal-5$^i$ in Fig. 6. The 3$^{\text{rd}}$ column of maps represents the self-similarity of the $F_{SAM}^t$. We introduce the $3 \times 3$ average pooling for $F_{SAM}^t$ followed by computing the cosine similarity between the pooled features and $F_{SAM}^t$. The maps illustrate that $F_{SAM}^t$ can accurately identify the regions of objects within the image, where the features within each object region are nearly identical, while features between neighboring different objects are distinct.

Although the characteristics of $F_{SAM}^t$ ensure the generation of high-quality masks, the coarse-grained features are not suitable for locating the objects, as shown in the 4$^{\text{th}}$ column. The similarity between $F_{SAM}^t$ and $F_{SAM}^r$ cannot effectively distinguish the target object well compared to $S_{mean}^+$ from DINOv2. Therefore, we follow the previous methods using DINOv2 for fine-grained features.

### A.5.2 Masks analysis for Point-Mask Clustering

Our Point-Mask Clustering module introduces a parameter-free clustering method by constructing a graph of coverage. The effectiveness of the method primarily relies on the high-quality masks, whose boundaries mostly align with the object boundaries. We roughly analyze the coverage of masks generated from the points in the ground truth foreground region using 4000 samples from Pascal-5$^i$. In particular, we get the union of masks from the foreground points as $\hat{y}_{fore}$, and visualize three distributions, including the distribution of IoU between $\hat{y}_{fore}$ and union of masks from background points $\hat{y}_{back}$ in Fig. 7, the ratio between the number of background points and all points covered by the $\hat{y}_{fore}$ in Fig. 8, the number of background points covered by $\hat{y}_{fore}$ in Fig. 9. The distribution charts demonstrate that most of the samples have acceptable coverage on the error points for Point-Mask Clustering. Given the limited precision of ground truth annotations, the analysis is for reference only. The effectiveness of our Point-Mask Clustering is validated in our ablation study in Tab. 4.

Reference Image   Target Image     Self Similarity   Cross Similarity     $S_{mean}^{+}$

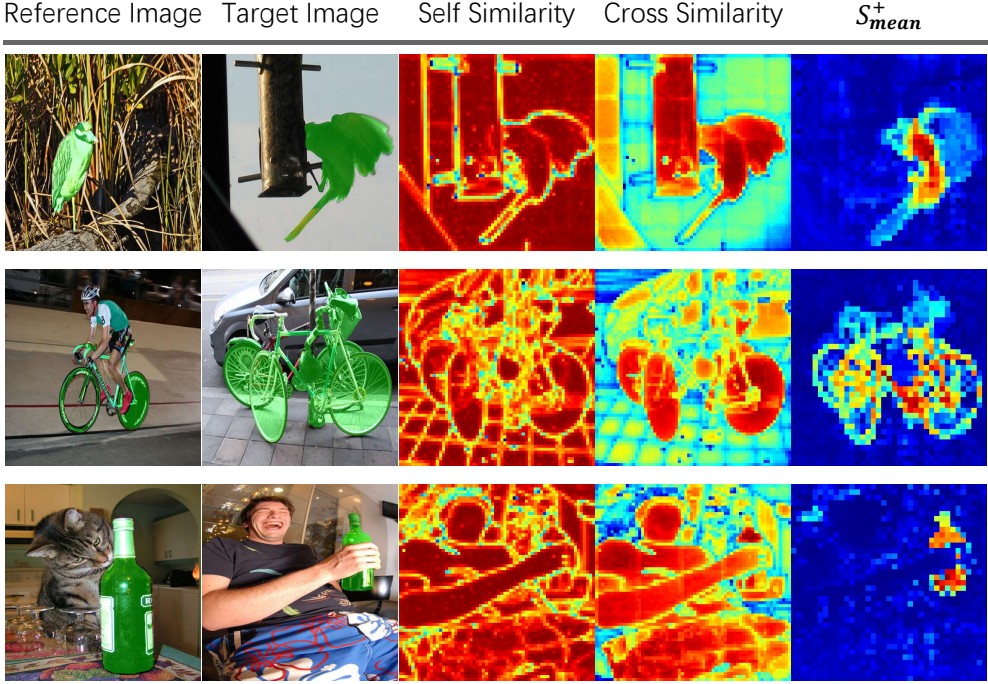

Figure 6: Analysis of the features from default ViT encoder of SAM.

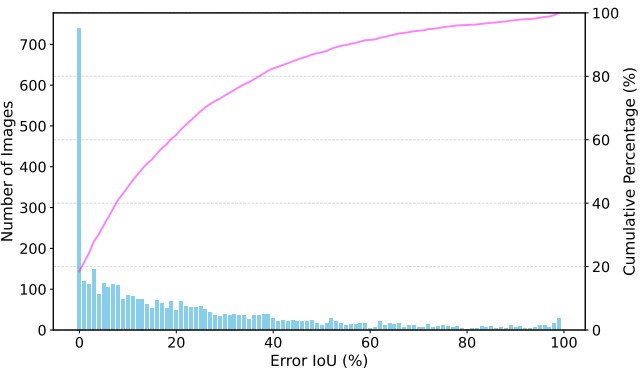

Figure 7: The distribution of IoU between masks from foreground points and from background points.

### A.6 Additional Experiment Results.

#### A.6.1 Performance of Different Foundation Model Sizes

Tab. 8 shows the experiment results of our approach with different scales of SAM and DINOv2. Compared to the previous training-free method Matcher, our approach still achieves a better performance with SAM-Large and DINOv2-Base. The fair comparison with SAM-Huge and DINOv2-Large further demonstrates the effectiveness of our approach.

#### A.6.2 Detailed Results of Evaluation Datasets

We present the detailed results on different Few-shot Semantic Segmentation datasets, including Pascal-5[i] in Tab. 9, COCO-20[i] in Tab. 10, LVIS-92[i] in Tab. 11, PASCAL-Part and PACO-Part in

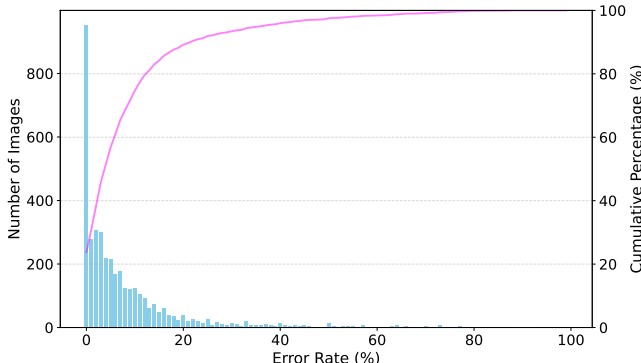

Figure 8: The distribution of the ratio between the number of background points and all points covered by the masks from foreground points.

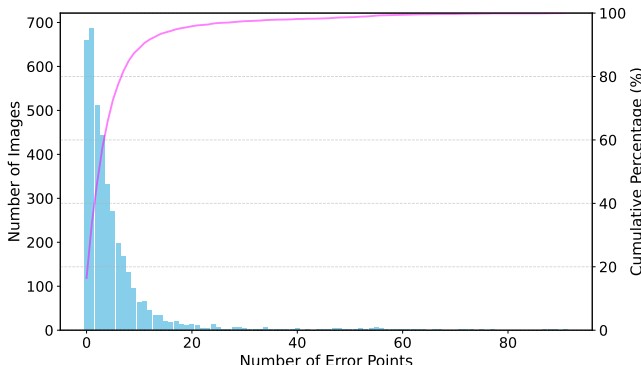

Figure 9: The distribution of the number of background points covered by the masks from foreground points.

Tab. 12, iSAID-5[i] in Tab. 13. The results show that our approach has remarkable performance in each fold of the datasets, demonstrating its generalized effectiveness in various scenarios.

### A.6.3   Multiple Random Seeds Experiment

Previous state-of-the-art methods, including both specialist methods and generalist methods, typically do not conduct multiple random seed experiments to evaluate the robustness. To demonstrate the robustness of our approach, we randomly set 5 different random seeds and conducted the experiments on the datasets that were not fully evaluated in the standard evaluation. As shown in Fig. 10, despite variations in random seeds, our approach consistently exhibits better performance compared to previous methods that were not evaluated with random seeds.

### A.7   Additional Ablation Study

### A.7.1   Ablation Study of Pivots for Positive Gating

We apply both $s_{mid}$ and $S_{mean}^-$ as the pivots for Positive Gating in Sec. 4.3, aiming to leverage both the pivots from the $S_{mean}^+$ itself and the negative similarity. As shown in Tab. 14, combining these two pivots for Positive Gating yields a significant improvement compared to using only one pivot. Moreover, the combination method of $\times$ shows a 0.3% mIoU enhancement compared to $+$.

Table 8: Evaluation of our approach with different sizes of SAM and DINOv2.

| Methods | SAM | DINOv2 | Params. | COCO-20$^i$ | FSS-1000 | LVIS-92$^i$ |
|---------|-----|--------|---------|----------|----------|----------|
| Matcher | huge | large | 945M | 52.7 | 87.0 | 31.4 |
| Ours | base | base | 180M | 53.7 | 85.6 | 31.1 |
| | large | base | 399M | 55.5 | 87.5 | 31.7 |
| | large | large | 617M | 58.0 | 87.8 | 35.1 |
| | huge | large | 945M | 58.7 | 88.0 | 35.2 |

Table 9: Detail results of Pascal-5$^i$.

| Methods | Pascal-5$^i$ 1-shot | | | | | Pascal-5$^i$ 5-shot | | | | |
|---------|-------|-------|-------|-------|------|-------|-------|-------|-------|------|
| | fold0 | fold1 | fold2 | fold3 | mean | fold0 | fold1 | fold2 | fold3 | mean |
| AMFormer | 71.3 | 76.7 | 70.7 | 63.9 | 70.7 | 74.4 | 78.5 | 74.3 | 67.2 | 73.6 |
| Matcher | 67.7 | 70.7 | 66.9 | 67.0 | 68.1 | 71.4 | 77.5 | 74.1 | 72.8 | 74.0 |
| Ours | 71.1 | 75.7 | 69.2 | 73.3 | 72.1 | 81.5 | 86.3 | 79.7 | 82.9 | 82.6 |

Table 10: Detail results of COCO-20$^i$.

| Methods | COCO-20$^i$ 1-shot | | | | | COCO-20$^i$ 5-shot | | | | |
|---------|-------|-------|-------|-------|------|-------|-------|-------|-------|------|
| | fold0 | fold1 | fold2 | fold3 | mean | fold0 | fold1 | fold2 | fold3 | mean |
| AMFormer | 44.9 | 55.8 | 52.7 | 50.6 | 51.0 | 52.0 | 61.9 | 57.4 | 57.9 | 57.3 |
| Matcher | 52.7 | 53.5 | 52.6 | 52.1 | 52.7 | 60.1 | 62.7 | 60.9 | 59.2 | 60.7 |
| Ours | 56.6 | 61.4 | 59.6 | 57.1 | 58.7 | 67.1 | 69.4 | 66.0 | 64.8 | 66.8 |

Table 11: Detail results of LVIS-92$^i$.

| Methods | LVIS-92$^i$ 1-shot | | | | | | | | | | |
|---------|-------|-------|-------|-------|-------|-------|-------|-------|-------|-------|------|
| | fold0 | fold1 | fold2 | fold3 | fold4 | fold5 | fold6 | fold7 | fold8 | fold9 | mean |
| Matcher | 31.4 | 30.9 | 33.7 | 38.1 | 30.5 | 32.5 | 35.9 | 34.2 | 33.0 | 29.7 | 31.4 |
| Ours | 30.9 | 37.9 | 37.1 | 39.6 | 31.2 | 36.4 | 39.1 | 35.7 | 32.3 | 31.5 | 35.2 |

| Methods | LVIS-92$^i$ 5-shot | | | | | | | | | | |
|---------|-------|-------|-------|-------|-------|-------|-------|-------|-------|-------|------|
| | fold0 | fold1 | fold2 | fold3 | fold4 | fold5 | fold6 | fold7 | fold8 | fold9 | mean |
| Matcher | 37.0 | 36.6 | 47.3 | 39.1 | 37.1 | 41.8 | 42.7 | 37.7 | 37.9 | 43.3 | 40.0 |
| Ours | 42.1 | 38.4 | 50.0 | 42.5 | 42.0 | 46.5 | 46.4 | 41.5 | 43.7 | 48.4 | 44.2 |

Table 12: Detail results of PASCAL-Part and PACO-Part.

| Methods | PASCAL-Part | | | | | PACO-Part | | | | |
|---------|---------|--------|--------|----------|------|-------|-------|-------|-------|------|
| | animals | indoor | person | vehicles | mean | fold0 | fold1 | fold2 | fold3 | mean |
| HSNet | 21.2 | 53.0 | 20.2 | 35.1 | 32.4 | 20.8 | 21.3 | 25.5 | 22.6 | 22.6 |
| Matcher | 37.1 | 56.3 | 32.4 | 45.7 | 42.9 | 32.7 | 35.6 | 36.5 | 34.1 | 34.7 |
| Ours | 33.2 | 59.6 | 35.2 | 50.1 | 44.5 | 33.4 | 34.9 | 39.7 | 37.0 | 36.3 |

Table 13: Detail results of iSAID-5$^i$.

| Methods | iSAID-5$^i$ 1-shot | | | | iSAID-5$^i$ 5-shot | | | |
|---------|-------|-------|-------|------|-------|-------|-------|------|
| | fold0 | fold1 | fold2 | mean | fold0 | fold1 | fold2 | mean |
| FRINet | 46.5 | 36.9 | 43.9 | 42.6 | 48.9 | 38.1 | 46.5 | 44.5 |
| Matcher | 37.3 | 23.8 | 38.8 | 33.3 | 38.3 | 24.0 | 40.6 | 34.3 |
| Ours | 53.4 | 36.8 | 51.2 | 47.1 | 59.3 | 39.9 | 58.0 | 52.4 |

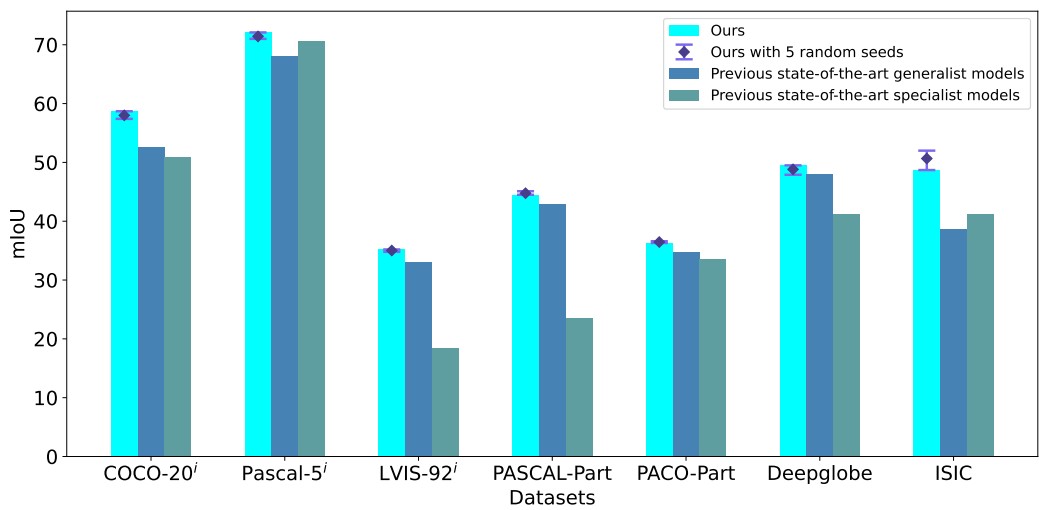

Figure 10: Results of our approach in multiple random seeds experiment. The bars in the chart represent the result under the previous standard evaluation. The error bar depicts the boundaries of our performance.

Table 14: Ablation study of pivots and combination operations in Positive Gating.

| Pivots | | Combination | COCO-20$^i$ |
|---|---|---|---|
| $s_{mid}$ | $S_{mean}^-$ | | |
| ✓ | | | 44.5 |
| | ✓ | | 51.0 |
| ✓ | ✓ | $+$ | 56.8 |
| ✓ | ✓ | $\times$ | 57.1 |

Table 15: Ablation study of different strategies for Positive Gating of the masks.

| Strategies | COCO-20$^i$ | LVIS-92$^i$ | PASCAL-Part | PACO-Part |
|---|---|---|---|---|
| Union | 59.4 | 36.1 | 40.1 | 31.6 |
| Mask Growth | 58.7 | 35.2 | 44.5 | 36.3 |

### A.7.2 Ablation Study of Other Strategies for Positive Gating

In Sec. 4.3 and Sec. A.1, we introduce the Mask Growth algorithm as our strategy for judging whether the mask is positive. We compare the strategy to separately judging each mask in Tab. 5 and judging the union mask of each cluster in Tab. 15. While simply judging the union mask shows better performance on COCO-20$^i$ and LVIS-92$^i$ that require complete coverage of objects, its performance on One-shot Part Segmentation has a significant decline. Considering the generalizability of our approach, we select the Mask Growth algorithm as our strategy.

### A.8 Additional Qualitative Analysis

We conduct additional qualitative analysis to better present the result of our approach. Fig. 11 further compare the Matcher, Baseline, B+PJ, and B+PJ+OJ (Ours) following Sec. 5.5. Fig. 12 illustrate the intermediate contents in the Post Gating. Moreover, we provide additional visualization results of standard FSS in Fig. 13, One-shot Part Segmentation in Fig. 14, and Cross Domain FSS in Fig. 15. These qualitative results demonstrate the effectiveness of our approach. Notably, some of the results are even better than the corresponding annotations.

Ref. Image    Tar. Image    Matcher    Baseline    B+PG    B+PG+OG(Ours)

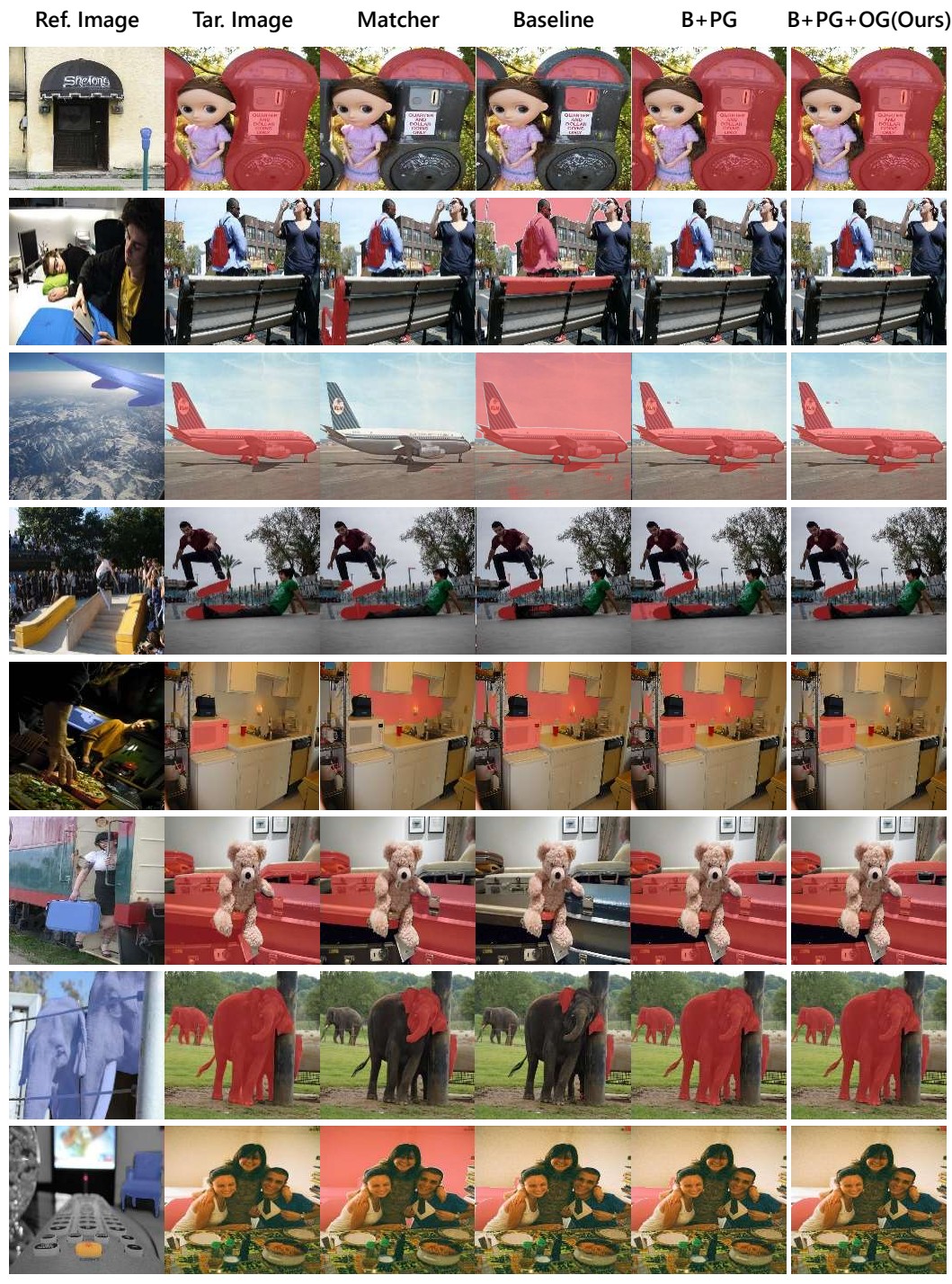

Figure 11: More Qualitative results on COCO-20[i] for comparison among Matcher, Baseline, B+PG, B+PG+OG.

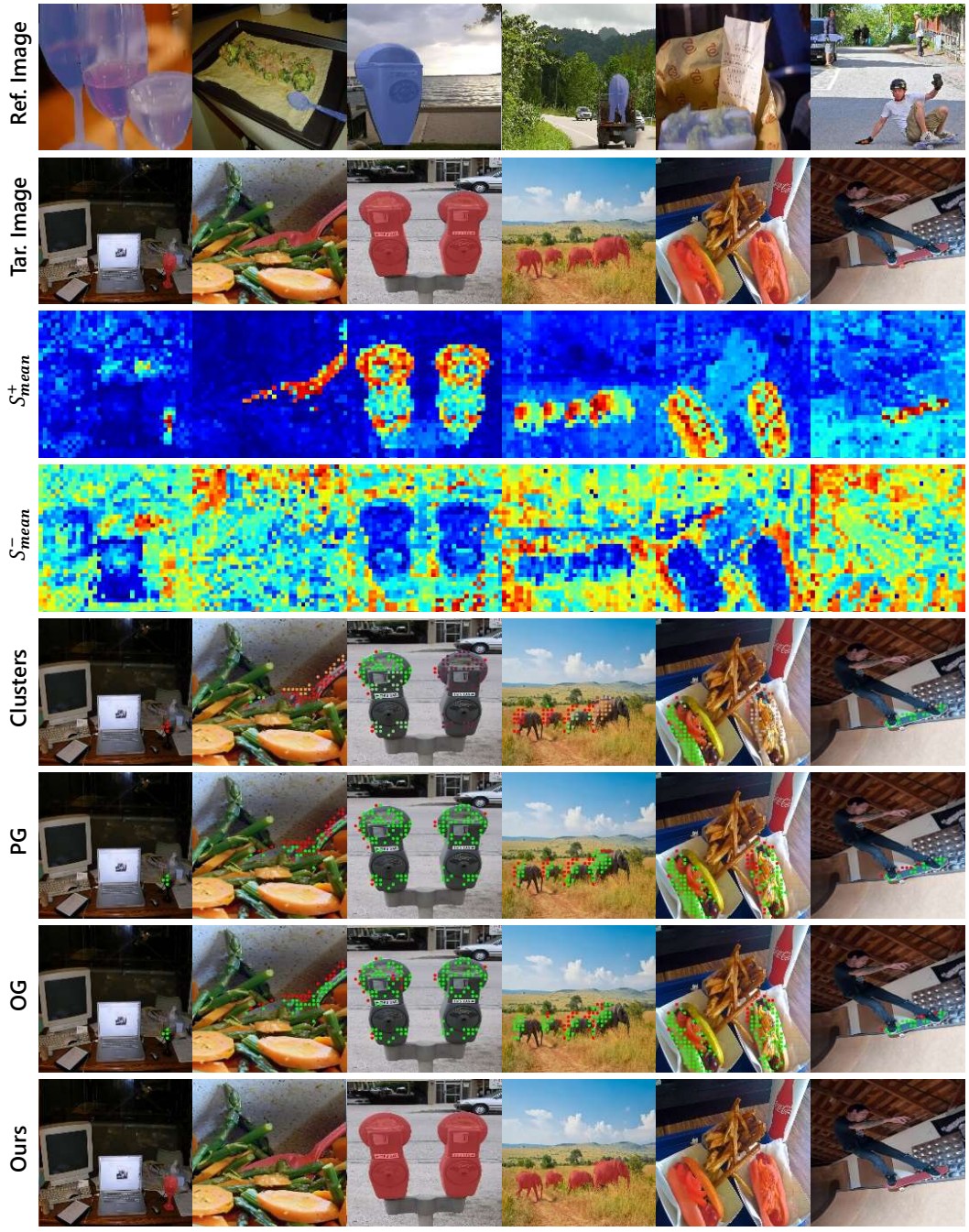

Figure 12: Qualitative analysis of the contents in gating. Different colors of points in the images in column "Clusters" represent different clusters. The green points in images in columns "PG" and "OG" denote the points satisfying the gating criteria, while the red points denote those not satisfying.

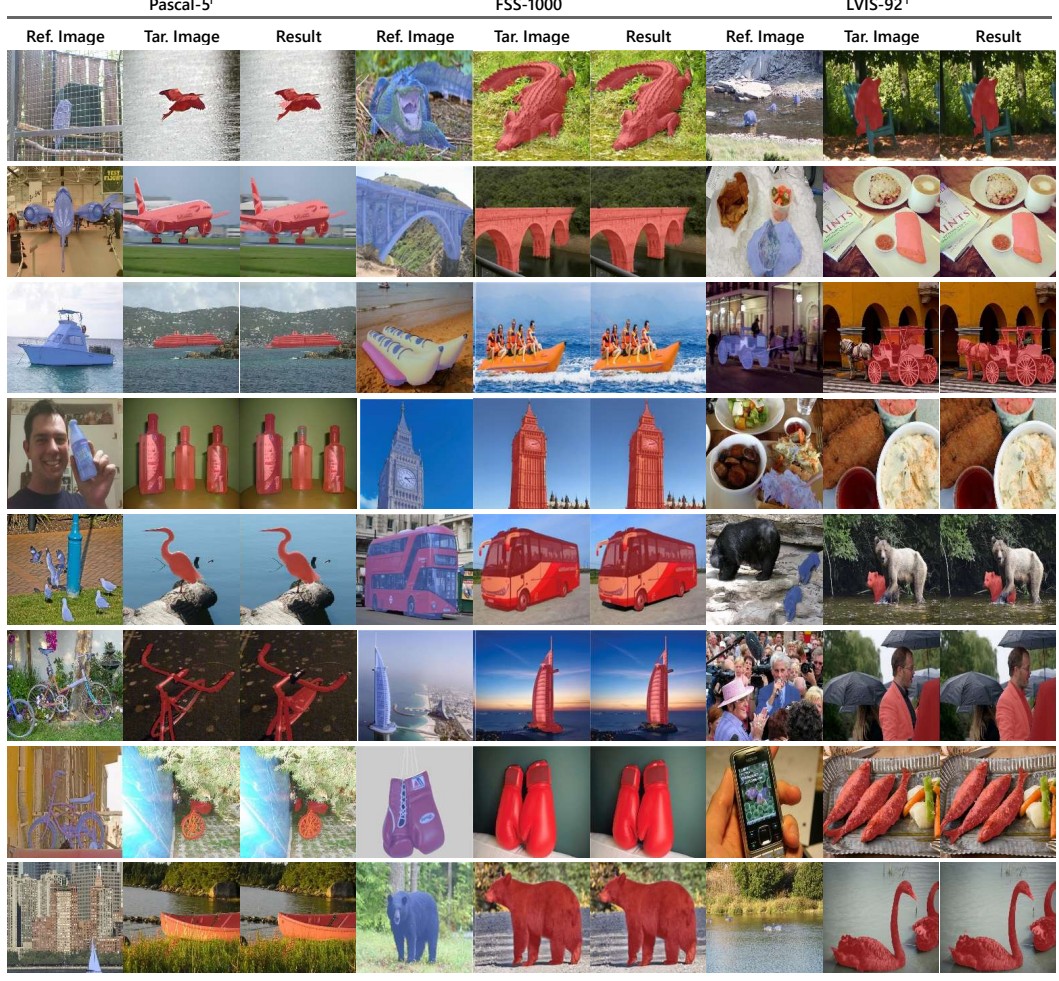

Figure 13: Qualitative analysis of the results on Pascal-5[i], FSS-1000, and LVIS-92[i].

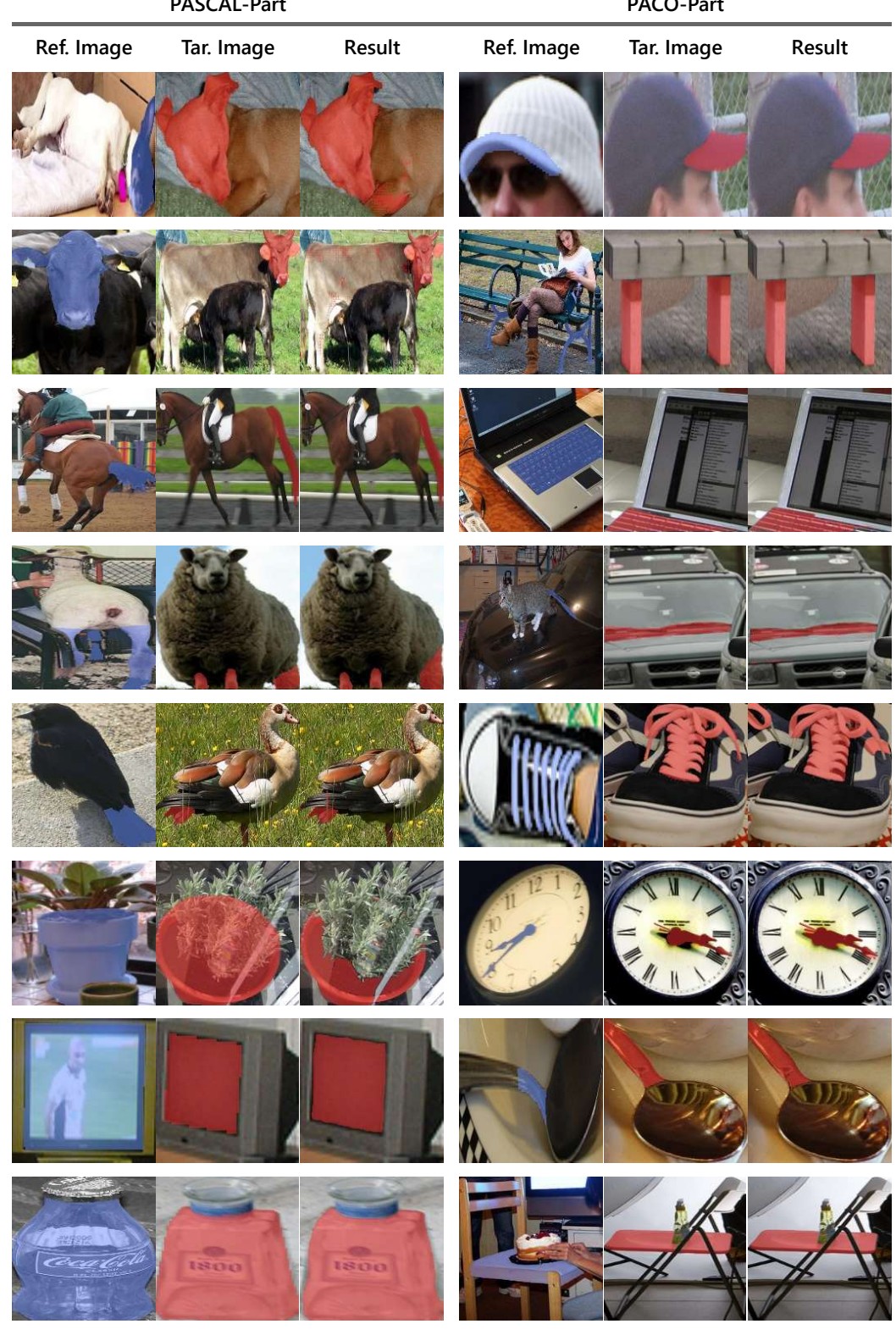

Figure 14: Qualitative analysis of the results on PASCAL-Part and PACO-Part.

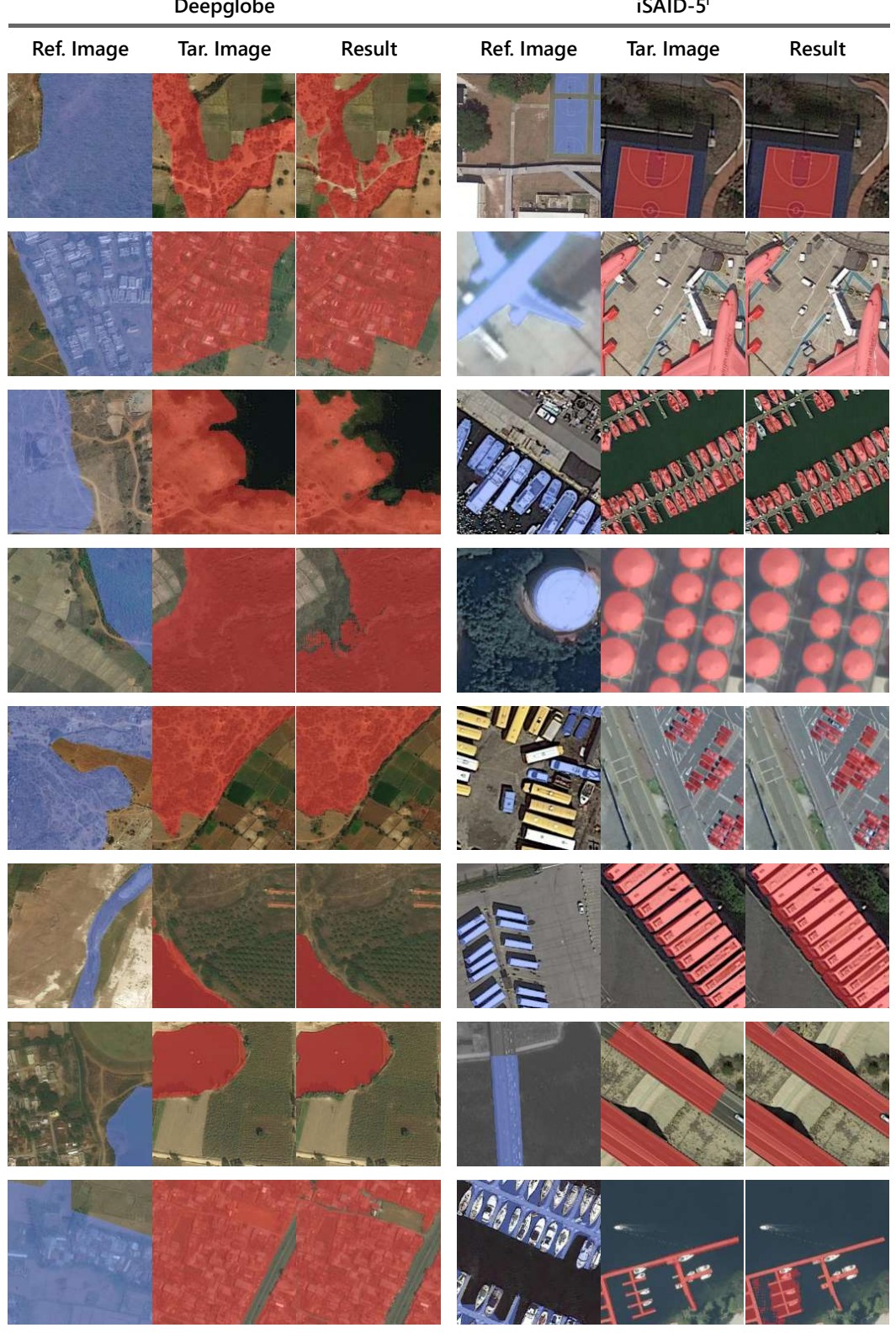

Figure 15: Qualitative analysis of the results on Deepglobe and iSAID-5[i].

