# OpenReview forum: "Bridge the Points: Graph-based Few-shot Segment Anything Semantically"
_NeurIPS.cc/2024/Conference — NeurIPS 2024 spotlight_

### Official Review · Reviewer_XvpJ · 2024-07-01

**Soundness:** 3
**Presentation:** 3
**Contribution:** 3
**Rating:** 7
**Confidence:** 5

**Summary:**

This paper proposes a graph-based approach for Few-shot Semantic Segmentation (FSS) based on the Segment Anything Model (SAM).
The authors propose a Positive-Negative Alignment module to select point prompts and a Point-Mask Clustering module to align the granularity of masks and selected points.
The proposed method surpasses state-of-the-art generalist models on COCO-20i and LVIS-92i datasets, and also performs well on One-shot Part Segmentation and Cross Domain FSS datasets.
Moreover, the proposed method is hyperparameter-free and efficient.

**Strengths:**

1. How to conduct prompt engineering for vision foundation models(SAM) is still a challenging problem in the research community. This work fills this gap.
2. The proposed method is systematic, consisting of three modules: Positive-Negative Alignment, Point-Mask Clustering, and Positive and Overshooting Gating. Each module is well-designed and novel.
3. The paper is well-written and easy to follow. The authors can accurately describe the well-designed modules.
4. The experiments are comprehensive and the results are convincing. The proposed method significantly outperforms the SOTA on COCO-20i. The cross-domain validation further demonstrates the generalization and robustness of the method.
5. The proposed method is hyperparameter-free, which is a great advantage and increases the practical value of the method.
6. I briefly checked the code, and the submitted code is complete, which increases the reproducibility of the paper.

**Weaknesses:**

1. In L173, "This is the precondition for the
efficacy of our PMC module, as even slight errors could significantly impact the clustering accuracy"
The authors should provide some visualization examples or experiments to verify this statement.

2. In Table 4, I notice weak or strong connected components are not clearly defined and the difference are also very small. The authors should provide more analysis to explain this.
And what is the hyperparameter for K-means++? Have you tried different hyperparameters for K-means++?

3. I notice the authors conduct ablation studies on different datasets for different components. I hope the authors can provide more experiments on part segmentation. Since previous SOTA Matcher require different hyperparameters for part segmentation. Conducting ablation on part segmentation can further prove the hyperparameter-free advantage of the proposed method.

**Questions:**

Can this methods applied to other tasks such as video object segmentation or instance segmentation?

**Limitations:**

The authors have discussed the limitations of the proposed method in appendix.

---

> ### Author Rebuttal · Authors · 2024-08-07
>
> # Response to weaknesses
>
> 1.
> >In L173, "This is the precondition for the efficacy of our PMC module, as even slight errors could significantly impact the clustering accuracy" The authors should provide some visualization examples or experiments to verify this statement.
>
> To verify this statement, we presented several experiments and visualizations. Specifically, 1) the visualization results in Fig.3 and Fig.12 illustrate the clustering results of the images. While the PNA module provides reliable point prompts, it is noted that some points fall *slightly outside* the target area. These misplacements may lead to generating masks for adjacent objects. Such points cannot be distinguished for clustering methods, particularly when the masks from SAM are of suboptimal quality or when there is **overlap** between masks of neighboring objects. 2) We further presented an analysis of the overlapping in Sec. A.5.2 and Fig. 7-9 in the Appendix (page 15), where it was observed that most adjacent masks along the boundary of foreground and background regions show limited overlapping. 3) According to Tab.8, our approach using a larger SAM model with better mask generation ability led to better results, showing the influence of mask quality.
>
> 2.
> >In Table 4, I notice weak or strong connected components are not clearly defined and the difference are also very small. The authors should provide more analysis to explain this.
>
> Weakly or strongly connected components are the concepts of graph connectivity, where each weak or strong connected component ensures that every vertex has connectivity or bidirectional paths from every other vertex in the directed graph, respectively. According to our visualization analysis in Fig. 18 of the global PDF, clustering by weak connected components prioritizes overall target coverage, while clustering by strongly connected components focuses on individual parts. Besides, we list the performance of these categories in the table below. Both of the clustering methods have their own advantage, where weak connected components can identify the well-shaped objects occupying enough proportion of the image area (shown in *Airplane* and *Elephant*), whereas the strong connected components can better handle the slim objects with limited pixels wide in the narrowest area (shown in *Scissors, Skateboard*). Therefore, by only using any of strong or weak connected components cannot address the two problems, and as a result, we have to select one method with better mIoU performance, i.e., the weak connected components.
>
> |Classes|Airplane|Elephant|Scissors|Skateboard|
> |-|-|-|-|-|
> |Ours w. strong|49.6|83.0|68.4|63.0|
> |Ours w. weak|65.0|84.0|68.0|59.3|
>
> 3.
> >And what is the hyperparameter for K-means++? Have you tried different hyperparameters for K-means++?
>
> Following the setting in Matcher [2], we set the hyperparameter K for K-means++ to 10, according to their ablation studies. We have also experimented with different hyperparameters and found that even if the K-means++ clustering method introduces an external hyperparameter, it does not surpass the performance of our mask clustering method (**58.7**).
> |K|6|8|10|
> |-|-|-|-|
> |Ours w. K-means++|57.7|57.8|57.5|
>
> 4.
> >I notice the authors conduct ablation studies on different datasets for different components. I hope the authors can provide more experiments on part segmentation.
>
> Thanks for the suggestion. We included experimental results on part segmentation in Table 2, as shown below. From these results, we can observe that the proposed method effectively handles part segmentation, validating the hyperparameter-free advantage of our approach compared to the Matcher which uses manually set hyperparameter. This indicates that our method not only simplifies the modeling process by eliminating the need for tuning external hyperparameters but also maintains robust performance across different segmentation tasks.
> |Method|PACO-Part|Pascal-Part|
> |-|-|-|
> |Matcher|34.7|42.9|
> |Ours|36.3 (+1.6%)|44.5 (+1.6%)|
>
> In this paper, we propose a Positive-Negative Alignment (PNA) module and a Post-Gating strategy based on the weakly connected graph components, enabling a hyperparameter-free pipeline.
> *As suggested, we further provide extensive ablation experiments on part segmentation detailed in the table below.* Our approach, employing various settings of parameter-free components, outperforms the previous SOTA, Matcher, which relies on different hyperparameters for part segmentation. We are happy to provide more ablation experiments if needed.
>
> |$S_{mean}^+$|$S_{max}^+$|$S_{mean}^-$|PACO-Part|Pascal-Part|
> |-|-|-|-|-|
> |$\checkmark$| |$\checkmark$|35.7|44.0|
> | |$\checkmark$|$\checkmark$|34.9|44.1|
> |$\checkmark$|$\checkmark$| |35.4|44.2|
> |$\checkmark$|$\checkmark$|$\checkmark$|**36.3**|**44.5**|
>
> # Response to questions
>
> 1.
> >Can this methods applied to other tasks such as video object segmentation or instance segmentation?
>
> Thanks for the insightful question. Our method follows the previous settings of FSS, i.e. segmenting one target class at a time. As a semantic segmentation method, our approach focuses on intra-class similarity and inter-class distinction. Whereas video object segmentation and instance segmentation require recognizing instance-level distinctions within the same class, which conflicts with the intra-class similarity emphasis of FSS. However, **as suggested**, we evaluate our approach on DAVIS 2017 via simple one-shot segmentation and argmax-based object classification using the mean similarity map from DINOv2, and achieved results that are 0.5% better than those of PerSAM (ICLR2024) under a similar VOS setting. Considering that the clustering process of our method is able to distinguish the instances to some extent, we will follow the suggestion from the reviewer and extend our method for these tasks in future work.

---

> > ### Comment · Reviewer_XvpJ · 2024-08-08
> >
> > After reading the author's response, most of my concerns have been addressed. I've decided to maintain my current score.

---

> > > ### Author Response · Authors · 2024-08-12
> > >
> > > We thank the reviewer for taking the time to review our response. We’re pleased that our rebuttal has addressed the concerns and we sincerely appreciate the reviewer's positive score. We will incorporate the key points the reviewer raised as we work on the final version of the revised paper.

---

### Official Review · Reviewer_zdFP · 2024-07-07

**Soundness:** 3
**Presentation:** 3
**Contribution:** 3
**Rating:** 6
**Confidence:** 4

**Summary:**

The paper first proposes graph-based approach for SAM-based few-shot semantic segmentation, modeling the relationship of SAM-generated masks in an automatic clustering manner. A positive-negative alignment module and a post-gating strategy based on the weakly connected graph components, enabling a hyper parameter-free pipeline. Extensive experimental comparisons and analysis across several datasets over various settings show the effectiveness and efficiency of the proposed method.

**Strengths:**

1、The graph-based approach for SAM-based methods proposed in the article is more insightful. Because SAM-based methods can generate many patches with point prompts, and how to group them is a good question.
2.The experiments are comprehensive. The ablation experiment demonstrates the effectiveness of the modules. The visualisation also illustrates the idea expressed.

**Weaknesses:**

1. The captions are not detailed, especially in Figure I and 2.
2. The overall organisation and presentation of the article are poor and need improvement.
Some details:
① On the left side of the first line of Figure 2, the images are ‘reference on top and target on bottom, but the features become ‘reference on bottom target on top’; it's not consistent.
② In my understanding, in Figure 2，positive-negative alignment should be conducted on the target image using the foreground and background features of the reference's image, but the part is very unclear and not informative and does not show the above content.
③Point-Mask Clustering part in Figure 2 can be understood that it wants to express the cluster of the points to different targets in the images, but the drawing is very casual, and there is no mark or expression.
3. How to conduct mask merging? Is each point used to generate a mask and then merge, or is the cluster of points used together to SAM to generate mask prediction, or are there other ways? Give the details.

**Questions:**

See the weakness part.

**Limitations:**

The author provides the limitations in the appendix. They say the coordinates of points will miss the small objects. This is a real problem and is suitable for further study.

---

> ### Author Rebuttal · Authors · 2024-08-07
>
> 1.
> >The captions are not detailed, especially in Figure I and 2.
>
> We apologize for the brevity of the captions due to consideration of space constraints. As suggested, we will include more details in the captions.
>
> Specifically, for Figure 1, the updated caption will be: "*Performance comparison of our approach against previous state-of-the-art methods in terms of efficiency and generalized capabilities in Few-shot Semantic Segmentation. Figure 1(a) illustrates our approach's superior performance in efficiency and effectiveness across various model sizes. Figure 1(b) demonstrates the generalizability of our approach across different domains.*"
>
> For Figure 2, the revised caption will be: "*Overview of our approach, where the Positive-Negative Alignment module recognizes the correlation between target features and reference features for point selection, the Point-Mask Clustering module efficiently clusters the points based on the coverage of corresponding masks, and Post-Gating filters out the false-positive masks for generating final prediction.*"
>
> 2.
> >The overall organisation and presentation of the article are poor and need improvement. Some details: ① On the left side of the first line of Figure 2, the images are ‘reference on top and target on bottom, but the features become ‘reference on bottom target on top’; it's not consistent. ② In my understanding, in Figure 2，positive-negative alignment should be conducted on the target image using the foreground and background features of the reference's image, but the part is very unclear and not informative and does not show the above content. ③Point-Mask Clustering part in Figure 2 can be understood that it wants to express the cluster of the points to different targets in the images, but the drawing is very casual, and there is no mark or expression.
>
> We appreciate the detailed suggestions for enhancing our presentation. As suggested, we have revised Fig. 2 and visualize the process of the PNA module. These updated figures are included in the **attached one-page global PDF file**. **We will incorporate this update in the final version.**
>
> 3.
> >How to conduct mask merging? Is each point used to generate a mask and then merge, or is the cluster of points used together to SAM to generate mask prediction, or are there other ways? Give the details.
>
> We acquired a set of points from the PNA module, each serving as a prompt for generating a corresponding mask, with the one-to-one correspondence where one point generates one mask. In the Post-Gating process, we select the points along with their corresponding masks for mask merging. For the final prediction, we simply used the union of the selected masks to create a merged mask. **We will update the more detailed description above in the final version.**

---

> > ### Comment · Reviewer_zdFP · 2024-08-13
> >
> > After reading the author's response, most of my concerns have been addressed. I've decided to raise my score.

---

### Official Review · Reviewer_DDH8 · 2024-07-12

**Soundness:** 3
**Presentation:** 3
**Contribution:** 3
**Rating:** 6
**Confidence:** 3

**Summary:**

This paper extends SAM to few-shot semantic segmentation tasks by proposing an approach based on graph analysis and representation learning. The contributions include a Positive-Negative Alignment module to generate the initial points prompt using DINOv2 features, as well as Point-Mask Clustering and Post Gating modules to filter the proper points based on the SAM generated masks. The proposed method exhibits efficiency and performance advantages over previous work such as PerSAM and Matcher.

**Strengths:**

(1)	The paper is well-written and easy to follow.

(2)	The proposed method is hyperparameter-free and shows significant improvements in both efficiency and performance.

(3)	Experiments on various datasets and tasks validated the generality of the proposed method.

**Weaknesses:**

(1)	The proposed method is specifically tailored for DINOv2 and SAM, is it possible to apply it to other foundation models? This could broaden the impact of the method.

(2)	In Figure 1(b), the scales of the axes are unclear. Are the starting points of the nine axes all 0?

(3)	On the FSS-1000 dataset, the performance of the proposed method is comparable to Matcher, which deserves a discussion. For example, what factors may have contributed to this result.

**Questions:**

See weakness.

**Limitations:**

The limitations have been discussed.

---

> ### Author Rebuttal · Authors · 2024-08-07
>
> 1.
> >The proposed method is specifically tailored for DINOv2 and SAM, is it possible to apply it to other foundation models? This could broaden the impact of the method.
>
> Thanks for the suggestion. Our approach can be easily applied to other foundation models, and as suggested, we further provide experimental analysis on this, as shown in the table below. Since SAM is almost unique prompt-based SOTA foundation model for segmentation, we only focus on evaluating different backbones here. Particularly, we assess various backbone networks from leading vision-related foundation models within our approach and the previous SOTA, Matcher, using the widely-used backbones of CLIP (ViT-L/14) and ImageNet-pretrained ResNet-50 on the $COCO-20^i$ dataset. The results in the table below show that our approach maintains competitive performance across different backbones, highlighting its effectiveness and adaptability.
> |Backbone|Matcher|Ours|
> |:-:|:-:|:-:|
> |DINOv2|52.7|58.7|
> |CLIP(ViT-L/14)|32.2|40.2|
> |ResNet-50|24.9|32.3|
>
> 2.
> >In Figure 1(b), the scales of the axes are unclear. Are the starting points of the nine axes all 0?
>
> Thanks for pointing this out. The starting points of the axes are not 0, as we intended to highlight the relative performance of our approach compared to previous SOTA methods. Since the current SOTA approaches are far from 0, setting the starting point at 0 would not distinctly present the advantage of our approach. Therefore, we customize the starting point to 40% of our performance level, enhancing visibility and comparison clarity.
>
> 3.
> >On the FSS-1000 dataset, the performance of the proposed method is comparable to Matcher, which deserves a discussion. For example, what factors may have contributed to this result.
>
> This dataset consists mainly of images with large, clearly defined target objects against simplistic backgrounds, which tends to normalize results across different segmentation methods. Such simplicity can obscure the unique strengths of various approaches. Our approach, equipped with the Positive-Negative Alignment (PNA) module and Positive Gating, is optimized for complex scenarios where background and target objects are closely interlinked. However, the simple backgrounds in FSS-1000 do not fully demonstrate our approach's capabilities, resulting in performance comparable to approaches like Matcher. This highlights the importance of diverse datasets in fully assessing the effectiveness of segmentation methods designed for complexity. In the final version, we will expand our experimental analysis to further clarify this.

---

### Author Rebuttal · Authors · 2024-08-07

We appreciate the detailed and constructive comments from the reviewers. For each of the concerns/questions, we have provided replies, revisions, and additional experiments accordingly, and included **a global one-page PDF (as attached below)** for figures mentioned in our response.

Please let us know if you have any further comments, and we are more than happy to participate in the discussion.

---

### Comment · Area_Chair_qdAB · 2024-08-11
**Author-reviewer discussion**

Dear reviewer DDH8 and zdFP,

Since the authors provided their responses, please read the responses, respond to them on in the discussion, and discuss points of disagreement if necessary by Aug 13.

Best regards,

AC

---

### Decision · Program_Chairs · 2024-09-25

**Decision:**

Accept (spotlight)

**Comment:**

All reviewers land on the positive side. They all enjoy the significance of the problem explored in the paper, the comprehensive and convincing experimental results as well as good practicability of the proposed method, i.e., it is hyperparameter-free. The AC suggests acceptance and encourages the authors to include the discussion in the rebuttal into the final version.